Phylogenetic and taxonomic revisions of Jurassic sea stars support a delayed evolutionary origin of the Asteriidae

Fau Marine faum@si.edu 1
Wright David F. 1 2 3
Ewin Timothy A.M. 4
Gale Andrew S. 4 5
Villier Loïc 6
1 Department of Paleobiology, National Museum of Natural History, Smithsonian Institution , Washington , DC , United States of America
2 Sam Noble Oklahoma Museum of Natural History, University of Oklahoma , Norman , OK , United States of America
3 School of Geosciences, University of Oklahoma , Norman , OK , United States of America
4 Science Group, Natural History Museum , London , United Kingdom
5 School of the Environment, Geography and Geological Sciences, University of Portsmouth , Portsmouth , United Kingdom
6 Centre de Recherche en Paléontologie—Paris, Sorbonne Université , Paris , France
Corbí Hugo
Electronic publication date: 2024 Oct 31
Publication date: 2024
Volume: 12
Electronic Location ID: e18169
Received 2024 Apr 24; Accepted 2024 Sep 3
Copyright: ©2024 Fau et al.
Copyright year: 2024
Copyright holder: Fau et al.
License: This is an open access article distributed under the terms of the Creative Commons Attribution License, which permits unrestricted use, distribution, reproduction and adaptation in any medium and for any purpose provided that it is properly attributed. For attribution, the original author(s), title, publication source (PeerJ) and either DOI or URL of the article must be cited.
License URL: https://creativecommons.org/licenses/by/4.0/

Keywords: Phylogeny, Jurassic, Asteroidea, Echinodermata, Forcipulatacea, Taxonomy, New taxon

Funding: Swiss National Science Foundation No P500PN_206858 Synthesys GB-TAF-6581 Marine Fau received financial support from the Swiss National Science Foundation (No P500PN_206858), and by the program Synthesys (GB-TAF-6581) for a visit of the Natural History Museum, London, in June 2017. The funders had no role in study design, data collection and analysis, decision to publish, or preparation of the manuscript.

==============================
Background

The superorder Forcipulatacea is a major clade of sea stars with approximately 400 extant species across three orders (Forcipulatida, Brisingida, Zorocallida). Over the past century, the systematics of Forcipulatacea have undergone multiple revisions by various authors, with some considering numerous families such as Asteriidae, Zoroasteridae, Pedicellasteridae, Stichasteridae, Heliasteridae, Labidiasteridae, and Neomorphasteridae, while others recognized only two families (i.e., Asteriidae and Zoroasteridae). Recent molecular analyses have shown the artificial nature of some of these groupings. Notably, four well-supported clades (Zorocallida, Brisingida, Stichasteridae, and Asteriidae) emerged from a synthesis of morphological and molecular evidence. The majority of extinct forcipulatacean species have been placed in modern families. However, many of these fossil species are in need of revision, especially those species placed within the Asteriidae, the largest of all forcipulatacean families.

Methods

In light of recent advancements in forcipulatacean systematics, we comprehensively reassess six well-preserved Jurassic forcipulatacean taxa, including the earliest crown-group members from the Hettangian (∼201.4 Ma), and also describe two new Jurassic genera, Forbesasterias gen. nov. and Marbleaster gen. nov. We assembled the largest and most comprehensive phylogenetic matrix for this group, sampling 42 fossil and extant forcipulatacean species for 120 morphological characters. To infer phylogenetic relationships and construct an evolutionary timeline for the diversification of major clades, we conducted a Bayesian tip-dating analysis incorporating the fossilized birth-death process. A total of 13 fossil species were sampled in our analysis, including six taxonomically revaluated herein, two recently reappraised species from the Jurassic, and five additional species from the Cretaceous and Miocene.

Results

Contrary to prior assumptions, our results indicate that none of the Jurassic taxa investigated belong to Asteriidae or any other modern families, and instead represent stem-forcipulatids. Furthermore, our phylogenetic results suggest that Asteriidae likely originated during the late Cretaceous. Our findings highlight a greater early diversity within the Forcipulatacea than previously presumed, challenging existing perceptions of the evolutionary history of this significant clade of marine invertebrates.

Introduction

The Forcipulatacea is one of the major clades within the Neoasteroidea, with about 400 described extant species (Mah & Blake, 2012). The crown-group Neoasteroidea is believed to have diversified after the Permo-Triassic mass extinction (Blake, 1987; Gale, 1987). However, the earliest unambiguous forcipulatacean fossils are dated from the Lower Jurassic (Blake, 1990), resulting in a major gap in the understanding of the early history of the group. The Triassic group Trichasterospida has been interpreted as stem forcipulataceans (Blake, 1987; Blake & Hagdorn, 2003; Blake, Bielert & Bielert, 2006), but recent discoveries have challenged the phylogenetic status of Trichasteropsida (Thuy, Hagdorn & Gale, 2017; Villier et al., 2018). The early history of the Forcipulatacea and its relationship with Triassic groups is important for understanding the history of the Neoasteroidea, because of the ongoing uncertainty concerning their phylogenetic position within the Neoasteroidea and relationships with Palaeozoic taxa (e.g., Blake, 1987; Gale, 1987; Blake & Hagdorn, 2003; Gale, 2011a; Mah & Foltz, 2011a; Mah & Foltz, 2011b; Mah & Blake, 2012; Blake & Mah, 2014; Linchangco et al., 2017). The interpretation of fossils and the timing of character appearances are central to the debate of clade definition and deep phylogenetic relationships.

The superorder Forcipulatacea comprises three orders: the Forcipulatida, (about 250 extant species); the Brisingida, (about 110 extant species) and the Zoroacallida (less than 40 extant species) (Mah & Blake, 2012). The systematics of the Forcipulatacea has changed multiple times over the course of the last century, with some authors recognizing many families (e.g., Brisingidae, Pedicellasteridae, Heliasteridae, Asteriidae, Zoroasteridae, and Stichasteridae in Perrier, 1894; Asteriidae, Zoroasteridae, Pedicellasteridae, Labidiasteridae, Neomorphasteridae, in Clark & Downey, 1992; Clark & Mah, 2001) while others accepted only two (Asteriidae, Zoroasteridae; e.g., Fisher, 1928).

The majority of extinct forcipulatid species have been placed in extant families. Most of these have been assigned within the Asteriidae (e.g., Hystrixasterias hettangiurnus Blake, 1990; Germanasterias amplipapularia Blake, 1990; Polarasterias Rousseau & Gale, 2018 (in Rousseau, Gale & Thuy, 2018); Savignaster villieri Gale, 2011b). including the monospecific subfamily Dermasterinae (containing the fossil genus Dermaster) (Hess, 1972). Outside of the Asteriidae, only three extinct species have been interpreted as Stichasteridae (Argoviaster occultus Hess, 1972; Pegaster stichos Blake & Peterson, 1993; and Atalopegaster gundersoni Blake & Guensburg, 2016) and one interpreted as a Pedicellasteridae (Afraster scalariformis Blake, Breton & Gofas, 1996). There are no extinct taxa known for the Heliasteridae, but exceptionally preserved specimens attributed to the extant species Heliaster microbrachius Xantus, 1860 have been reported from the Pliocene of Florida, USA (Jones & Portell, 1988). More generally, Cenozoic fossils are rare and have all been compared to extant genera (Barker & Zullo, 1980; Blake & Zinsmeister, 1988; Blake & Aronson, 1998; Palópolo et al., 2021). The Brisingida are very rare in the fossil record, and are only known from the Miocene of Japan (Yamaoka, 1987).

Recent phylogenetic analyses allowed reappraisal of the historic classifications (e.g., Fisher, 1928; Fisher, 1930; Clark & Downey, 1992) of the Forcipulatacea (Mah, 2000; Foltz et al., 2007; Mah & Foltz, 2011a; Mah et al., 2015). The most comprehensive molecular-based phylogenetic analysis of Mah & Foltz (2011a) identified 4 main clades: Asteriidae, Stichasteridae, Zoroasteridae and Brisingida. They further suggested that the family Pedicellasteridae is polyphyletic, that Labidiasteridae is a synonym of Heliasteridae, and that Neomorphasteridae is a synonym of Stichasteridae (Mah & Foltz, 2011a). More recently, a study based on morphological data by Fau & Villier (2020) found congruent evidence supporting the same four major extant subclades within the Forcipulatacea as Mah & Foltz (2011a). In a second study focused on extant Zoroasteridae and their fossil relatives, Fau & Villier (2023) suggested the Mesozoic Terminasteridae are not monophyletic, instead arguing to resurrect and expand the Order Zorocallida Downey, 1970 to include both the crown group Zoroasteridae and fossil stem-group taxa. These recent phylogenetic analyses have greatly restricted the definition of the Asteriidae and suggest greater phylogenetic and taxonomic diversity within the Forcipulatacea than currently expressed in the literature (Mah & Foltz, 2011a; Mah et al., 2015; Fau & Villier, 2020).

To better understand the origin and early diversification of this major superorder, we present a taxonomic re-evaluation of six well-preserved Jurassic forcipulatacean species, including the earliest crown-group members. Further, we describe two new genera: Forbesasterias gaveyi gen. nov. (Forbes, 1850) and Marbleaster spiniger gen. nov. (Wright, 1880) and redescribe Argoviaster occultus Hess, 1972 and Dermaster boehmi De Loriol, 1899. Using the largest morphology-based character matrix ever constructed for the Forcipulatacea (Fau & Villier, 2020), we apply Bayesian tip-dating phylogenetic methods (Warnock & Wright, 2020; Wright, Wagner & Wright, 2021) to simultaneously co-estimate phylogenetic relationships and divergence times among fossil and extant Forcipulatacea. In addition to the six species revised herein, we also sampled seven other fossil forcipulatacean species (from Jurassic to Miocene) in our tip-dating analysis to assist the estimation of divergence times. Finally, the results of our divergence dating analysis allows us to investigate questions surrounding the origination and diversification of major forcipulatacean subclades, including the species-rich Asteriidae.

Material and Methods

Taxon sampling

The analysis focuses on the most completely known extinct Jurassic taxa to ensure the maximum number of characters could be scored. The six Jurassic taxa reappraised are represented by 37 specimens (see material examined under systematic palaeontology). The phylogenetic analysis is based on our reappraised descriptions and taxonomic revision. Seven extinct species were added in the phylogenetic analysis in order to obtain a more comprehensive temporal coverage of the Forcipulatacea. Two recently reappraised Jurassic taxa: Psammaster davidsoni (De Loriol & Pellat, 1874) reappraised in Fau et al. (2020); Terminaster cancriformis (Quenstedt, 1876) reappraised in Fau & Villier (2023); and three Cretaceous taxa: Pegaster stichos Blake & Peterson, 1993; Cretasterias reticulatus Gale & Villier, 2013; Viridisaster guerangeri Fau & Villier, 2023; and two Cenozoic taxa: Zoroaster marambioensis Palópolo et al., 2021; Brisingella sp.

Terminology

The anatomical descriptions follow the terminology outlined by Fau & Villier (2018) and Fau & Villier (2020). Anatomical terms and abbreviations from Fau & Villier (2018) and Fau & Villier (2020) are in italics in the text. We use conventional terms for the orientation of the specimen: abactinal (adoral) versus actinal (oral); proximal versus distal; and abradial versus adradial. In the literature, the size of an individual is commonly given with the two measures “r” and “R”, with r corresponding to the distance between the centre of the disc and the edge of the disc, and R corresponding to the distance between the centre of the disc and the tip of the arm.

Nomenclatural act

The electronic version of this article in Portable Document Format (PDF) will represent a published work according to the International Commission on Zoological Nomenclature (ICZN), and hence the new names contained in the electronic version are effectively published under that Code from the electronic edition alone. This published work and the nomenclatural acts it contains have been registered in ZooBank, the online registration system for the ICZN. The ZooBank LSIDs (Life Science Identifiers) can be resolved and the associated information viewed through any standard web browser by appending the LSID to the prefix http://zoobank.org/. The LSID for this publication is: [urn:lsid:zoobank.org:pub:6A43BD80-6C00-42C6-AFD4-7C4A944396FD]. The online version of this work is archived and available from the following digital repositories: PeerJ, PubMed Central SCIE and CLOCKSS.

Phylogenetic analysis

We expanded the morphological character matrix of Fau & Villier (2020) and Fau & Villier (2023) to sample a total of 13 fossil species (6 taxonomically re-evaluated herein) and 29 extant species for 120 characters (character matrix available in Supplementary Materials). The extant Plutonaster bifrons and Dactylosaster cylindricus were also sampled in our matrix as outgroup taxa. In this study, the character/taxon matrix of Fau & Villier (2020) and Fau & Villier (2023) was revised as followed: the character states of character 50 (number of primary spines per adambulacrals; character number 49 in Fau & Villier, 2020) were modified, as many extinct taxa possess three or four adambulacral spines (i.e., state 0: 1 to 2 adambulacral spines; state 1: three to four adambulacral spines; state 2: 5 and more adambulacral spines), and two characters were added (character 119 and 120; see Supplementary Materials File). The modified character by taxon matrix was coded using MESQUITE software (Maddison & Maddison, 2023). Our matrix is scored up to 78% complete (22% of missing or non-applicable characters), among which the 13 extinct taxa are scored in average up to 46.3% (median 48.3% complete), with Cretasterias reticulatus being the most complete (63.3%/76 characters scored) and Brisingella sp. the least complete (7.5%/nine characters scored). All characters can potentially be observed in extinct taxa depending on their preservation. However, some characters are more difficult to observe, such as characters of the oral frame (characters 4 to 25). Characters 4 to 17 were only scored/observed on Argoviaster occultus. In total, only 13 characters were not observed in any of the 13 extinct taxa studied (characters 18 to 24, 44 to 47 and 109 and 110).

To simultaneously co-estimate phylogenetic relationships and estimate divergences times, we conducted a tip-dated Bayesian phylogenetic analysis incorporating the fossilized birth-death process (FBD) (Stadler, 2010; Heath, Huelsenbeck & Stadler, 2014; Gavryushkina et al., 2014; Wright, 2017a; Warnock & Wright, 2020; Wright, Wagner & Wright, 2021). Bayesian phylogenetic methods using FBD models leverage both morphological and stratigraphic age information from the fossil record (Barido-Sottani et al., 2020; Wright, Wagner & Wright, 2021), which can then be combined with data from extant taxa to generate time-calibrated phylogenies containing both fossil and extant species. Tip-dating approaches using the FBD process provide a more coherent framework for dating lineage divergences than node-based approaches (Heath, Huelsenbeck & Stadler, 2014), and have also been shown to improve phylogenetic inferences involving fossil taxa compared to undated approaches (Barido-Sottani et al., 2020; Mongiardino Koch, Garwood & Parry, 2021). Moreover, Bayesian tip-dating approaches can be used to directly test macroevolutionary hypotheses about character evolution (Wright, 2017b; Wright, Wagner & Wright, 2021), patterns of clade diversification (Paterson, Edgecombe & Lee, 2019), and investigate the evolutionary origin of major clades (Wright & Toom, 2017; Thuy et al., 2022).

We applied the sampled-ancestor implementation of the FBD model (Gavryushkina et al., 2014), and placed broad, uniform priors on FBD parameters for diversification, extinction, and fossil sampling. Fossil ages were assigned uniform distributions based on their occurrences in geologic stages. An abundance of paleontological evidence points to a post-Permian origin of the Neoasteroidea (e.g., Blake, 1987; Gale, 1987; Villier et al., 2018). To incorporate this information while allowing for possible alternatives, we placed a prior distribution on the tree age that spans the Permo-Triassic boundary (∼240 Ma–260 Ma). Morphological character evolution was modelled using a variant of the simple Mk model (Lewis, 2001) that accounts for ascertainment bias and allows for morphological rates to vary among characters according to a lognormal distribution (Wagner, 2012; Wright, 2017a). To account for rate variation among lineages throughout the tree, we applied an uncorrelated morphological clock where branch rates vary according to an independent gamma rates model (Lepage et al., 2007).

To assist the analysis, we applied a topological constraint to the taxon Brisingella sp., a taxon so poorly preserved that only nine characters could be scored. Brisingella sp. possesses clear characters showing its affinities with the Brisingida, such as a large number of arms (9 arms), the presence of a rigid circular oral ring and the shape of its ambulacral (robust and hourglass). We followed the work of Zhang et al. (2024), which considers Brinsigella sp. as the sister taxa to the Freyellidae. Although poorly preserved, Brinsigella sp. is the only fossil of Brinsigida known to date and was included here to sample fossil representatives for all forcipulataceans orders.

Bayesian inference of phylogeny was estimated using Markov chain Monte Carlo (MCMC) simulation in MrBayes 3.2.6 (Ronquist et al., 2012). Two MCMC runs with four chains were run for 80,000,000 generations. Chains were sampled every 500 generations and the first 25% sampled were discarded as burn-in. Chains reached an average deviation split frequency of less than 0.01. Convergence was assessed using Tracer 1.7.1 (Rambaut et al., 2018), parameters attained effective sample sizes (ESS) >100, with all but one >1,000, and potential scale reduction factors (PSRF) of ∼1.0. The character/taxon matrix and MrBayes script are available in the Supplementary Materials.

Systematic paleontology

ASTEROIDEA De Blainville, 1830	
FORCIPULATACEA Blake, 1987	
Forbesasterias gen. nov.	
urn:lsid:zoobank.org:act:B14659DD-7EBD-411B-9471-54E6D9458288	

Type species. Uraster gaveyi Forbes, 1850.

Derivation of name. In honor of Edward Forbes, who described this specimen in 1850.

Diagnosis. As for species, by monotypy.

Forbesasterias gaveyi (Forbes, 1850)	
Figs. 1A, 2 and 3	
 	
1850Uraster gaveyiForbes decade III pl. II	
1854Uraster gaveyi,Forbes p. 90	
1863Uraster gaveyi, Wright, p. 100–102, pl. 1, Figs. 1A–1B	
1870Uraster gaveyi, Wright, p. 163	
1876Uraster graveyi, Quenstedt, p. 85, pl. 93, Fig. 29	
? 1935Asterias gaveyi, Mercier, p. 47, pl. 2, Figs. 18A–18B	
1966Asterias? gaveyi, Spencer & Wright p. U75, Fig. 66.1	
1972Asterias? gaveyi, Hess p. 32	
1993 “Asterias” gaveyi, Lewis, p. 48	
1996 “Asterias” gaveyi, Blake, p. 179	
2011 “Asterias” gaveyi, Gale p. 58, text-Figs. 24A–24C	

Type specimen. NHMUK PI E 1638, holotype.

Figure 1 Forbesasterias gaveyi gen. nov. holotype NHMUK PI E 1638 (A) and Marbleaster spiniger gen. nov. holotype NHMUK PI E 1642 (B).

Photos by Marine Fau. Scale bars: two cm.

Figure 2 Arms of Forbesasterias gaveyi gen. nov. (holotype, NHMUK PI E 1638).

(A) Close up on distal portion of an arm showing ambulacral, adambulacrals and body wall plates still bearing spines and aligned transversally. (B) Close up on actinals without spines and adambulacrals, proximal direction on the right. (C) Interbrachial area showing oral ossicles, adoral carina and giant straight pedicellariae (white circle). Abbreviations: adamb: adambulacrals; amb: ambulacrals. Photos by Marine Fau. Scale bars: five mm.

Figure 3 Forbesasterias gaveyi gen. nov. (holotype, NHMUK PI E 1638).

(A) Ambulacral furrow showing ambulacrals and adambulacrals, proximal direction to the right. (B) Close up on ambulacrals, proximal direction to the left. (C) Close up photograph of the giant straight pedicellariae in Fig. 2C, proximal direction to the bottom. (D) Drawing of the giant straight pedicellariae showing basal piece and valves. (E) Regular size straight pedicellariae (white circle) found around the ambulacral groove and ambulacrals, ambulacral groove to the top. Abbreviations: actam, transverse actinal interambulacral muscle; adamb, adambulacrals; amb, ambulacrals; amb base, ambulacral base; spa, spine attachment structures. Photos by Marine Fau. Scale bars: five mm (A); two mm (B–E).

Type locality. Mickleton tunnel, near Chipping Campden, Gloucestershire, England; Capricornus zone, Pliensbachian, Early Jurassic (Forbes, 1850; Wright, 1863).

Material examined. NHMUK PI E 1638, holotype.

Diagnosis. Forcipulatacean sea star with five arms, compressed ambulacrals and adambulacrals, and quadriserial ambulacral pores. Adoral carina composed of four elongated adambulacrals, bearing one or two spines each. Other adambulacrals bearing four spines. Actinal plates present. Straight pedicellariae present, regular straight pedicellaria approximately 1 or two mm long, and giant strait pedicellaria approximately five mm long. Giant straight pedicellariae located around the mouth in interbrachial areas. Abactinal spines short and numerous with ornamented tips.

Description. The holotype exposes the actinal face of a large individual, R >99 mm, r = 26 mm, with four arms preserved (Fig. 1). The specimen was slightly flattened during preservation processes. The ambulacral grooves are probably wider than they would have been in life. Some portions of the arms are more damaged than others, with disarticulated ambulacral heads in some part of the arms. Ambulacral grooves are less flattened and therefore narrower around the mouth frame. The disc and the structure of the wall skeleton are not exposed.

Ambulacrals are compressed lengthwise. The head is slightly broader than the shaft and the teeth are present along the entire width of the ambulacral head (Figs. 2A–2B, 3A–3B). The actam and the furrow are well defined (Fig. 3B), no wings on the ambulacral bases. Four tube feet rows per ambulacral groove. The distalmost part of one of the arms is crushed, allowing observation of the proximal side of a few ambulacrals and adambulacrals. The most distal ambulacrals are rather straight, their actinal edge being straight to slightly concave, as in many Forcipulatacea. Arching of the actinal edge of proximal ambulacrals is not visible due to preservation.

Adambulacrals are compressed lengthwise, bearing 4 spines each, arranged in a transverse row. The adambulacral spines are short and thick, slightly flattened at the extremity, but longer than the actinal spines. The spines do not seem to have glassy trabeculae or any ornamentation, instead are composed of undifferentiated labyrinthic stereom.

The adoral carina is composed of at least 4 elongate adambulacrals per ambulacral side (Fig. 2). Adambulacrals of the adoral carina have a triangular shape, whereas they are more or less square in others . They are also, at least, 1.5 times longer than the other adambulacrals with the two most proximal adambulacrals being the longest. There are only one or two spines on the adambulacrals of the adoral carina. The oral ossicles are short and bear at least two spines each.

Actinal and possibly marginal plates are visible, most of them still bearing spines. Actinals overlapping each other. Actinal rows can be distinguished from marginal rows by ending before the terminal ossicle. There are at least two rows of actinals, very likely more, but it is impossible to count actinal rows near the interradius where the number is usually at its maximum. The longest actinal row ends at least six millimetres before the end of the longest preserved arm tip, even though the actual tip is missing. This means that the actinals were an important part of the body wall. Actinal plates are small and stout. They are arranged to form regular lateral and longitudinal rows. Some actinals have a small central psas (i.e., primary spine attachment structure, also called “pustule” in Fau & Villier, 2020) supporting a primary spine, but this structure is not visible on every plate.

A row of bigger, slightly triangular plates, is visible in the interradial area. These plates are morphologically differentiated compared to the actinal plates and are here assumed to be marginal plates. Morphological differences between actinal and marginal plates reduces distally along the arm. Due to preservation, it is impossible to determine if a second marginal series is present. Abactinals and carinals cannot be observed. Short and slender spines, with ornamented extremities, may represent spines of dorsal plate series.

Giant straight pedicellariae (up to five mm long) are found proximally in the interradial space around the mouth and inside the ambulacral groove (Figs. 2C, 3C–3E). The giant straight pedicellariae are made of two slender and long valves that broaden at the base. Smaller straight pedicellariae, with two slender regular valves that are slightly flattened at the extremity, occur more distally along the arms. No crossed pedicellariae are recognized.

Remarks. The identification of the specimen NHMUK PI E 3339 as “Uraster” gaveyi is uncertain because numerous morphological differences are apparent with the holotype. The description is therefore based on the holotype only.

Mercier (1935) attributed disarticulated body wall skeleton ossicles materials from the Sinemurian of Normandy, France to “Asterias” gaveyi. However, this material cannot be reliably compared with the holotype and is therefore excluded.

The distinctive giant pedicellariae of Forbesasterias gaveyi gen. nov. are noticeable, but giant pedicellariae also occur in modern forcipulataceans taxa. For instance, the asteriid Notasterias armata possesses giant crossed pedicellariae on its abactinal surface. Large straight pedicellariae are also found in Zoroasteridae.

Forbesasterias gaveyi gen. nov.is clearly distinguished from other extant species of the genus Asterias by the lack of crossed pedicellaria, that it shows no evidence of intermarginals, and possesses four spines per adambulacral. Species of extant Asterias possesses both straight and crossed pedicellariae, clearly distinguishable intermarginals and adambulacrals with one to three spines.

Marbleaster gen. nov.	
urn:lsid:zoobank.org:act:6CB3E261-2BA3-48DB-A17C-9E7B3586A431	

Type species. Marbleaster spiniger (Wright, 1880)

Derivation of name. For the Forest Marble Formation.

Diagnosis. As for species, by monotypy.

Marbleaster spiniger (Wright, 1880)	
Figs. 1B and 4	
	
1880Uraster spinigerWright p. 166–167, Fig. 1; pl. XXI, Fig. 1	
1966Compsaster spiniger, Spencer & Wright p. U74, Figs. 65, 1C	
1993Compsaster spiniger, Lewis, p. 60	

Type specimen. NHMUK PI E 1642, holotype.

Figure 4 Marbleaster spiniger gen. nov. (holotype, NHMUK PI E 1642), in actual view.

(A) Oral frame and two arms with spines. (B) Oral frame showing the orals and adoral carina. (C) Straight pedicellariae (white circle) and crossed pedicellariae (white rectangle), proximal direction to the right, adradial direction to the bottom. (D) Close up on ambulacrals showing large ambulacral heads. Abbreviations: ad car, adoral carina; adamb, adambulacrals; amb, ambulacrals. Photos by Marine Fau. Scale bars: five mm (A); two mm (B–D).

Type locality. Near Rode (formerly Road), Somerset, United Kingdom; Forest Marble Formation, Bathonian, Middle Jurassic (Wright, 1880). Wright (1880) refers to the locality as “near Road, Wilts”, however the village of Rode sits on the Wiltshire to Somerset boarder and is now regarded as part of the latter county.

Material examined. NHMUK PI E 1642, holotype.

Diagnosis. Forcipulatacean sea star with five short arms. At least three spines per oral ossicle, short adoral carina composed of one or two adambulacrals only. Ambulacrals and adambulacrals compressed, three to four spines per adambulacral. Ambulacrals with an extended crest on the ambulacral’s head, and a well-defined furrow on the ambulacral’s shaft. Body wall plates present with keyhole-shaped primary spine attachment structure (psas) in the interradial area of the disc. Body wall ossicles bearing many long and slender spines, made of glassy trabeculae. Straight and crossed forcipulate pedicellariae of the same size sparsely distributed across the body.

Description. The specimen has a diameter of about four cm. Only the actinal surface is visible. The body is flattened, two arms are broken, and most of the ambulacral grooves are covered by sediments. The body wall skeleton is not accessible. Some plates in the interradial area can be interpreted as actinal plates.

The oral frame is characterised by five pairs of long and narrow oral ossicles, each bearing at least one spine directed proximally and one or two spines oriented actinally. Short adoral carina composed of the first proximal adambulacral only, with some second adambulacrals in contact but not compressed in width. Adambulacrals of the adoral carina are narrower and longer than other adambulacrals. They possess only one psas, instead of three as the rest of the adambulacrals along the arms.

Ambulacral grooves are partially covered by spines and sediments. Ambulacrals are compressed in length, the head larger than the shaft, slightly hourglass-shaped. The actam and the furrow (on the ambulacral shaft) are well defined. No wings on the ambulacral bases. A crest, similar to the ambulacral crest of extant asteriids (Fig. 4; Fau & Villier, 2020) is present on the head and it is tilted in a proximal direction.

The adambulacrals bear three to four relatively thick and long spines (two mm) with ornamented tips and attached to a psas. The adambulacrals are irregular in size, highly compressed in length, and wider than high.

The body wall plates are present in the interradial area. At least 2 rows of body wall plates can be identified. They are likely to be actinal plates because of their small size and placement. One of these plates possesses a keyhole-shaped psas. The arm plates are covered by long and conic spines made of glassy trabeculae.

Both straight, duck-billed pedicellariae and crossed forcipulate pedicellariae are present (Fig. 4C). They are randomly distributed between the spines across the surface of the specimen. Straight and crossed pedicellariae are of similar size (one mm). The crossed pedicellariae are relatively large, some are as long as half the length of the spines. Crossed pedicellariae are similar in shape to those of modern Stichasteridae and Asteridae (e.g., Fisher, 1928; Fisher, 1930; Clark & Downey, 1992), and are randomly distributed on the abactinal surface. There is no evidence of wreath organs/cluster of crossed pedicellariae.

Remarks. “Uraster” spiniger was originally classified within the asteriids by Wright (1880). However, he expressed doubt about the systematic position of his new species: “This starfish differs so much from the other fossil species of the genus Uraster that it may possibly prove to be the type of a new genus, when more details are learned anent the anatomy of the skeleton by the discovery of new materials” (Wright, 1880, p. 167). Spencer & Wright (1966) assigned “Uraster” spiniger to the genus Compsaster. However, the type species of the genus Compsaster formosus Worthen & Miller, 1883 from the Carboniferous of Illinois differs in many aspects from “Uraster” spiniger Wright, 1880. These differences have been outlined by Blake (2002, p. 363): “Although the type specimen of the Jurassic species Compsaster spiniger is imperfectly preserved, it appears readily assigned to the surviving Asteriidae, a family known from the beginning of the Jurassic (Blake, 1990), well before the Bathonian occurrence of C. spiniger. The Compsasteridae therefore here is restricted to the type species”. As outline by Wright (1880), Marbleaster spiniger gen. nov. exhibits distinctive characters absent in other Jurassic species that warrant the establishment of a novel genus to house this species. These include the presence keyhole-shaped psas, and the presence of straight and crossed pedicellariae of equal size uniformly distributed across its actinal surface.

Dermaster boehmi De Loriol, 1899	
Figure 5	
	
1899Dermaster boehmiDe Loriol, p.1–6, pl. 1, Fig. 1	
1972Dermaster boehmi, Hess p. 32–36, text-Fig. 3, 31–39, 89; pl. 2, Fig. 1, 3; pl. 3, Fig. 1; pl. 4, Fig. 1; pl. 12, Fig. 2	
1973Dermaster boehmi, Hess, p. 627	
2011 Dermaster boehmi, Gale p. 60	

Type specimen: Specimen illustrated by De Loriol, 1899, pl. 1, Fig. 1 (assumed lost).

Figure 5 Arrangement of the body wall and arm ossicles of Dermaster boehmi, Photographs (A, C) and interpretation drawings (B, D).

(A–B) NMB M10678. (C–D) NMB M10705. Dash lines indicates uncertain contour of the ossicles, ? indicates uncertainty on ossicle homology. Abbreviations: interrad; primary interradial; im, inferomarginal; sm, superomarginal; rad, primary radial. Coloured areas indicate ossicle homology. In orange, primary central ossicle; in grey, disc abactinals; in red, madreporite; in dark blue, primary interradials; in dark green, primary radials; green, carinals; in pink, abactinals; in blue, superomarginals; in yellow, inferomarginals; in brown, actinals; in purple, adambulacrals; in teal, spines. Photos by Marine Fau. Scale bars: five mm.

Type locality. Vögisheim, Mülheim, Baden-Württemberg, Germany; ferruginous layers in limestone, Bathonian, Middle Jurassic (De Loriol, 1899). From the information provided by De Loriol (1899; locality, age and geology), the holotype is assumed to come from the ferruginous oolitic layer of the Hauptrogenstein-Formation (Bloos, Dietl & Schweigert, 2006).

Material examined. Six well-preserved specimens described by Hess (1972): NMB M8985, M10678, M9365, M9600, M10705, and M9168. All specimens originate from the village of Schinznach, Canton of Aargau, Switzerland and were collected from the Upper Hauptrogenstein-Formation (Upper Bajocian; Middle Jurassic).

Diagnosis (emended from Hess, 1972). Aboral skeleton reticulate, pore fields present but small. Body wall ossicles cruciform to triangular, covered by small granule-like spines. One or two psas on primary interradials and primary radials. Primary radials overlapping the primary interradials. Adoral carina present, composed of the two most proximal adambulacrals. Ambulacrals compressed, ambulacral pores biserial. Adambulacrals with three to four spines each. Straight and crossed pedicellariae present. Crossed pedicellariae differentiated into two morphotypes.

Description. The disc is composed of five primary radials and five primary interradials that are arranged around a circlet of small abactinals and the primary central plate (Figs. 5A–5B). The primary central plate is approximal twice the size of the disc abactinals. The superomarginal plate rows extend into the disc to join and to partially overlap the primary interradials. The most proximal superomarginals of adjacent arms are in contact inter-radially on the disc. In NMB M10678, only one of the enlarged primary radials overlaps the primary interradials directly. The other four primary radials are slightly set distally and not in contact with the other primary interradials. All disc plates are covered by little bumps, that indicate the former presence of spines. NMB M10705 is an arm fragment, with what is likely to be a few plates from the disc. One primary radial is visible and bears a relatively big psas in its center. The radial of NMB M10705 is twice the size of those of NMB M10678 and clearly show two types of attachment for the spines, a big central psas and some smaller bumps around the psas, as already described by Hess (1972). At least two types of spines (i.e., primary spines attached on psas, and secondary spines attached on bumps) are present. The madreporite is not preserved in any of the specimens studied. In NMB M10678, a cavity remains in one of the primary interradials, where the madreporite inserted (Figs. 4.5A–4.5B). De Loriol (1899) and Hess (1972) described the madreporite as a small swollen plate, that “lies near the edge of the interbrachial angle” (translation of De Loriol, 1899 p. 5). Current observations and the description of De Loriol (1899) and Hess (1972) agree with the conclusion that the madreporite was not fused with an interradial. A strong relationship between one of the interradials and the madreporite is possible, as in the Zoroasteridae in which the madreporite sets in a special cavity on the distal edge of an interradial.

The arms are composed of at least nine plates rows in addition to the adambulacral and ambulacral columns: one row of carinals, one row of abactinals, two rows of marginals, and one or two rows of actinals, on each side. Proximally, carinals, abactinals and superomarginals are cruciform, but the articular processes get shorter distally (Figs. 5B and 5D). All plate surfaces are granulated. The carinal row is regular along the arms. The carinals were formerly adorned by many small spines, and at least one big stout rounded spine. Spines are better preserved in NMB M10705: there are big and stout, blunt primary spines, and small acicular secondary spines. There is no doubt that there were many secondary spines per plates because of their granulated surfaces. Because of the number of primary spines preserved, it is also likely that each plate was bearing one or more primary spines. However, psas are not observed on every plate.

Abactinals are small and irregular in shape. They are overlapped by both the carinals and the superomarginals. Superomarginals are cruciform proximally, but of rather heterogenous shape distally. Inferomarginals are smaller than the superomarginals. All bore many spines, at least one primary spine and many secondary spines. The terminal ossicle is round and relatively big, with a granular surface, probably bearing many spines too. At least one row of actinals is present on the specimens studied. The actinals are small, with a cruciform to triangular shape, the abactinal lobe tends to be reduced.

Ambulacrals are compressed, but not as much as in modern Asteriidae. Tube feet are arranged in two rows in the ambulacral groove. The head of the ambulacrals are slightly longer than the shaft, but symmetrical, the furrow on the shaft is well marked, and no wings on the ambulacral bases. Adambulacrals are also compressed and bear a transverse row of at least three, maybe four spines per adambulacral. Adambulacral spines are conical, long and slender. They are the longest spines present. These long and slender spines are preserved only around the adambulacrals. There are no signs of long primary spines on the actinals or on the inferomarginals. The adoral carina is short and composed of only the two most proximal adambulacrals.

Both straight and cross pedicellariae are preserved in NMB M10705 and M10678. Straight pedicellariae are similar to straight pedicellariae of extant taxa and are the most visible on the actinal surface of NMB M10678-B. It is possible to recognize two types of crossed pedicellariae. In NMB M10678, small crossed pedicellariae are abundant, especially between around the marginals and the actinals. In NMB M10705, on the other hand, crossed pedicellariae are larger and more robust.

Remarks. Dermaster boehmi seems to present two different types of crossed pedicellariae. Only the “robust” crossed pedicellariae were described and illustrated by Hess (1972, Figs. 33, 35–37). D. boehmi is not the only species to present different morphotypes of pedicellariae. For instance, Pisaster ochraceous has two types of straight pedicellariae while Pedicellaster hypernotius has two types of crossed pedicellariae (Fau & Villier, 2020). Even if this is rare among the Forcipulatacea, this is not a unique case of multi-pedicellariae morphotypes. However, until this polymorphism is found in other fossil taxa, this should be regarded as an autapomorphy of D. boehmi.

Argoviaster occultus Hess, 1972	
Figure 6	
1972Argoviaster occultusHess p. 27–32, text-Figs. 29–30; pl. 9, Fig. 2; pl. 10–11; pl. 12, Fig. 1; pl. 13, Fig. 1; pl. 14, Fig. 1	

Type specimen. NMB M8977, holotype.

Figure 6 Arrangement of the body wall and arm ossicles of Argoviaster occultus, Photographs (A, C, E) and interpretation drawings (B, D).

(A–B) NMB M9366. (C–D) NMB M9362. (E) NMB M8977, the holotype. Dash lines indicates uncertain contour of the ossicles. Abbreviations: amb, ambulacral; adamb, adambulacral; im, inferomarginal; sm, superomarginal. Coloured areas indicate ossicle homology. In orange, primary central ossicle; in grey, disc abactinals; in red, madreporite; in dark blue, primary interradials; in dark green, primary radials; green, carinals; in pink, abactinals; in blue, superomarginals; in yellow, inferomarginals; in brown, actinals; in grey, ambulacrals; in purple, adambulacrals. Photos by Marine Fau. Scale bars: five cm (A–B), one cm (C–E).

Type locality. Schinznach, Canton of Aargau, Switzerland; Upper Hauptrogenstein-Formation, Upper Bajocian, Middle Jurassic (Hess, 1972).

Material examined. NMB M8977, holotype; NMB M9359, NMB M9362, NMB M9366 NMB M9344/1-2, NMB M9360/1-2, NMB M9361/1-2, NMB NMB M9364/1-2, NMB M10676, paratypes; NMB M9460, NMB M9465, NMB M9469, NMB M9475, NMB M9479, NMB M9480/1-2, NMB M9481, NMB M9482, NMB M9483, NMB M9487, NMB M9489, NMB M9505, NMB M9506/1-2, NMB M9514/1-2.

Diagnosis (emended from Hess, 1972). Forcipulatacean sea star with five arms. Ambulacrals and adambulacrals compressed. Ambulacral pores quadriserial. Adambulacral with three short spines each. Plates of the arms arranged in longitudinal and transverse rows, with a small papular field at each corner. Body wall ossicles triangular to cruciform, with reduced ornamentation of granules, sometimes with one central psas, and occasionally a central cavity. Straight and crossed forcipulate pedicellariae present.

Description. The holotype, NMB M8977 is a distal part of an arm, showing mostly the ambulacral groove. NMB M9366 is composed of four arms partially preserved in abactinal view, and a few ossicles of the fifth arm still embedded in the matrix. NMB M9362 has five partially preserved arms, the body wall skeleton is mostly missing, so that the oral frame and the ambulacral skeleton can be observed from the inside.

The structure of the wall skeleton is visible on one arm of NMB M9366, but because the specimen is flattened, the arms appear larger than they would have been in life. All the skeleton arm plates in NMB M9366 are small and triangular, the surface is finely granulated with no psas. The central plates row is assumed to be the carinal row (Figs. 6A–6B), because of its central position, and because the carinals overlap their abactinal neighbours on each side of the arm. On each side of the carinals, there is at least one row of abactinals, with small plates intercalated in between the abactinal and carinal rows. All arm plates series are similar in shape and size and it is difficult to differentiate abactinal plates from the marginal plates. There are possibly one row of superomarginals and one row of inferomarginals on each side of the arm. The second-best preserved arm of NMB M9366 shows part of the carinal row proximally, overlapping some abactinals. The rest of the carinals and abactinals have been removed, exposing the ambulacrals, adambulacrals and some actinals and maybe inferomarginals that are cut transversally (Figs. 6A–6B). There are at least three or four rows of actinals.

Wall skeleton plates in NMB M9362 look different in shape compared to the triangular plates observed in NMB M9366, but the differences could be explained by the different views offered by the two specimens. Ambulacrals and adambulacrals are easily recognizable in NMB M9362 (Figs. 6C–6D), but the wall skeleton plates are more difficult to recognise because they were scattered by taphonomic disarticulation. Hess (1972) recognized difficulties in the identification of superomarginals in M9362. Assuming that the wall skeleton plates of A. occultus follow the Forcipulatid Plating Rules (Gale, 2011a), the homologies of the actinal, inferomarginal, superomarginal, abactinal and carinal series can be recognized from their relative position, shape and number of articulation areas. As the body wall plates overlap one another, they present a number of articulation areas on their external and internal faces. When looking at the internal faces, carinals and superomarginals should exhibit three articulation areas, the inferomarginals only two, and the actinals and abactinals one or two. Carinals are cruciform, abactinals seems to be triangular in radial cross-section, or rod-like in actinal view. Superomarginals are also cruciform, but higher than long, with a well-developed actinal lobe. Inferomarginals on the contrary have a more developed abactinal lobe. Actinals seems to be more or less cruciform. Two rows of actinals are visible in the holotype. They strongly overlap each other and bear one psas per plate (Fig. 6E). All three specimens have wall skeleton plates with a central cavity, which is a unique feature in the Forcipulatacea.

Ambulacrals are highly compressed and gently curved to accommodate four alternate tube feet rows in the ambulacral groove. The ambulacral crest is tilted proximally. There is no wing on the ambulacral bases. A furrow is present and well-marked on the ambulacral shaft. The adambulacrals are compressed as well and bear three spines each.

The specimen M9362 partially exhibits the oral frame ossicles. Only the ramus of the orals is visible. Several spines are preserved in the center of the circle formed by the oral frame, but it is not possible to count or estimate the number of spines per orals. First ambulacrals have a shape similar to modern Asteriidae with a long head, and a long, well-developed, but not high, proximal process. The odontophores are square and were probably connected to both the orals and first ambulacrals with the articulation areas poda and doda clearly separated. The crater seems to be present.

Pedicellariae were not found in NMB M9362. In NMB M9366, Hess (1972) described remains of pedicellariae, scattered between the body wall ossicles. In the holotype, both straight and crossed forcipulate pedicellariae are present along the ambulacral groove (Fig. 4.7).

Remarks. There is no mention of crossed pedicellaria in the original description of the species by Hess (1972), but they are present at least in the holotype. In addition, the present description provides more detail about the structure of the body wall skeleton (Fig. 6).

Germanasterias amplipapularia Blake, 1990	
Figure 7	
1990Germanasterias amplipapulariaBlake p. 103–123, Figs. 1–2	
2011 Germanasterias amplipapularia Gale p. 57	

Type specimen. SMNS 18869a–b.

Figure 7 Artificial cast of Germanasterias amplipapularia, holotype Nr. 18869a–b in abactinal view (A), and actinal view (B). (C) Details of the disc and arm in abactinal view. (D) Detail of the arms and oral frame in actinal view.

Abbreviations: adamb, adambulacral; ad car, adoral carina; mad, madreporite; sp, straight pedicellariae. Photos by Andrew S. Gale. Scale bars: five cm (A–B), two mm (C–D).

Type locality. Göppingen, Baden-Württemberg, Germany; Schlotheimia angulata Zone, late Hettangian, Early Jurassic (Blake, 1990).

Material examined. SMNS 18869a–b, holotype.

Diagnosis (emended from Blake, 1990). Forcipulatacean with body wall skeleton consisting of a carinal series, one row of abactinals (and associated smaller plates) on each side of the carinal series, two marginal and three actinal series. Carinals and marginals alternating between spine bearing and non-spine bearing plates. Only one large primary spine per carinal or marginal with spines. Abactinals digitate, arranged in regular transverse and longitudinal rows and separated by relatively large papular area. Ambulacrals compressed, podial pores quadriserial. Adoral carina composed of the first four to five adambulacrals. Adambulacrals compressed, bearing four spines each. Straight duck billed pedicellariae present on abactinal surface only, very abundant.

Remarks. For complete description, see Blake (1990). Blake (1990) described the adambulacrals of Germanasterias amplipapularia as weakly carinate proximally. In the literature, adambulacrals of forcipulatacean sea stars are described as carinate if they possess an adradial extension (Fau & Villier, 2018). Alternate carinate and non carinate adambulacrals are a synapomorphy of the Zoroasteridae (Fau & Villier, 2020). Some other forcipulatacean taxa (e.g., Heliaster, Asterias) can have alternating sized adambulacrals, with generally the larger sized adambulacrals bearing one more spine than the smaller sized adambulacrals. However different sized adambulacrals cannot be considered homologous with the adradial extension of the Zoroasteridae. The scoring of G. amplipapularia for phylogenetic study follows the scoring system of Fau & Villier (2020), considering carinate adambulacrals absent on both G. amplipapularia and H. hettangiurnus.

Hystrixasterias hettangiurnus Blake, 1990	
Figure 8	
1990Hystrixasterias hettangiurnusBlake p. 103–123, Figs. 3–4	
2011 Hystrixasterias hettangiurnus Gale p. 57–58, text-Figs. 24D–24G	

Type specimens. NMB M9682, holotype; NMB M9681, NMB M9684-8, paratypes.

Figure 8 Hystrixasterias hettangiurnus, paratype NMB M9681 (A–B), holotype NMB M9682 (C–D).

(A) Abactinal view of NMB M9681. (B) Details of the disc of NMB M9681. (C) “duck-bill” pedicellariae, NMB M9682. (D) “robust clam shell” pedicellariae, NMB M9682. Abbreviations: mad, madreporite; sp, straight pedicellariae; 1st amb, 1st ambulacral. Photos by Marine Fau. Scale bars: one cm (A–B), one mm (C–D).

Type locality. Schechingen, Baden-Württemberg, Germany; Hettangian (Formation unknown), Early Jurassic (Blake, 1990).

Material examined. NMB M9682, holotype; NMB M9684, NMB M9686

NMB M9687, NMB M9688, paratypes. Specimen NMB M9685 was missing from the NMB collections, and assume to be lost.

Diagnosis (emended from Blake, 1990). Forcipulatacean sea star with body wall skeleton consisting of carinals bordered on each side by up to three rows of abactinals, two rows of marginals and two to three rows of actinals. Abactinals arranged in regular transverse and longitudinal rows. Ambulacral moderately compressed, podial pores quadriserial. Adoral carina composed of the first three adambulacrals. Adambulacrals with transverse series of four prominent spines. Straight forcipulate pedicellariae present. Straight pedicellariae differentiated in two types: clam-shaped pedicellariae present on abactinal surface only and duck billed pedicellariae present on actinal surface.

Remarks. For complete description, see Blake (1990).

Hystrixasterias hettangiurnus possess two different morphotypes of straight pedicellariae, relatively robust clam shell pedicellariae as illustrated by Gale (2011a), text-Figs. 24D–24G), and modern-like duck billed pedicellariae (Figs. 8C–8D). H. hettangiurnus lacks keyhole-shaped psas (synapomorphy of the Asteriidae) and possess a madreporite that is not fused with an interradial (synapomorphy shared by the Asteriidae and the Stichasteridae).

Blake (1990) described the adambulacrals of H. hettangiurnus and Germanasterias amplipapularia as weakly carinate respectively medially and proximally. Adambulacrals of H. hettangiurnus are not here considerate carinate (see remarks for Germanasterias amplipapularia).

Results

The main clades recovered in both the maximum credibility clade (MCC) tree (Fig. 9) and the 50% majority rule consensus (MRC) tree (Fig. 10) are: Brisingida, Zorocallida, Zoroasteridae, Heliasteridae, Stichasteridae and Asteriidae (Fau & Villier, 2020; Fau & Villier, 2023). Terminaster cancriformis is found to be sister taxa to the Zoroasteridae in both the MCC tree and the majority rule consensus tree, retrieving the clade Zorocallida which is consistent with the results in Fau & Villier (2023). The position of Hystrixasterias hettangiurnus, Germanasterias amplipapularia, Forbesasterias gaveyi gen. nov., Dermaster boehmi and Psammaster davidsoni are found to be uncertain, but deeply rooted at the base of the Forcipulatida clade (Fig. 10). The position of P. davidsoni is compatible with the previous phylogenetic analysis sampling this taxon by Fau et al. (2020). Dermaster boehmi and Psammaster davidsoni are well supported as sister taxa (posterior probability = 0.75) (Fig. 10). Our results do not support F. gaveyi, G. amplipapularia and H. hettangiurnus as members of the family Asteriidae. Argoviaster occultus and Marbleaster spiniger are found to be the most derived of the Jurassic taxa reappraised here, as sister taxa to all extant forcipulatids (Figs. 9 and 10).

Figure 9 Maximum clade credibility tree of the Forcipulatacea including 13 extinct taxa.

Posterior probabilities are shown at each nodes, blue node bars represent the 95% highest posterior density age estimates. Major clades are indicated by black dots. Stratigraphic ranges are indicated by thick black bars.

Figure 10 Fifty percent majority rule consensus tree summarizing posterior distribution of trees resulting from Bayesian analyses; posterior probabilities are shown at each nodes.

Major clades are indicated by black dots. Stratigraphic ranges are indicated by thick black bars.

The Forcipulatida is composed of the clade Asteriidae + Stichasteridae, the Heliasteridae, the extant species Pedicellaster hypernotius and the extinct taxa Argoviaster occultus, Marbleaster spiniger and Pegaster stichos (Fig. 10). Although the phylogenetic positions of these taxa are poorly supported (Figs. 9 and 10), our results indicate none of the three extinct species belong to any extant families. Both P. stichos and A. occultus have been compared to the extant Neomorphaster and placed in the subfamily Neomorphasterinae (now synonymized with Stichasteridae; Hess, 1972; Blake & Peterson, 1993). However, our results do not support either of them as stichasterids. Cretasterias reticulatus Gale & Villier, 2013 is found to be the sister taxa to the clade formed by all extant asteriids. This position is compatible with either its inclusion within the family Asteriidae or to be interpretated as a stem-Asteriidae. Cretasterias reticulatus possesses 4 out of the 6 synapomorphies proposed by Fau & Villier (2020): char. 28 ambulacral with strongly arched abactinal profile; char. 29 the muscle insertion lim represent more than 40% of the ossicle height and finishing under the actam; char. 68. Round psas absent on the inferomarginals; char. 111. Wreath organ present. Wreath organs are clusters of crossed pedicellariae arranged around primary spines of asteriids, and are capable of moving along the spines in response to stimuli (Lambert, De Vos & Jangoux, 1984). However, C. reticulatus does not have any differentiated abactinals, which are the last two synapomorphies of the extant family (char. 80 abactinals differentiate with at least two level of plates; and char. 82 abactinals, intercalary inter-arc ossicles present).

Discussion

The Asteriidae (Gray, 1840) was the first named family in the Forcipulatacea, initially including all known forcipulataceans. The species assigned to this family have greatly changed over time, as it was progressively split into several families. Fisher (1928) and Fisher (1930) placed many species into the family Asteriidae, including taxa that are currently classified in the families Pedicellasteridae and Stichasteridae. Fisher’s classification was followed by many authors (e.g., Clark & Downey, 1992; Mah, 2000), until phylogenetic analysis, based on molecular data, radically changed the circumscription of the Asteriidae (Mah & Foltz, 2011a). Mah & Foltz’s (2011a) phylogenetic hypothesis supports a restricted definition of the Asteriidae. The Asteriidae and the Stichasteridae were retrieved as two distinct clades, and the family Pedicellasteridae as polyphyletic (Mah & Foltz, 2011a). Even in its current and restricted definition, the Asteriidae is still the most diverse family of all living forcipulatacean, representing half of the species diversity of the group (Mah & Blake, 2012). The phylogenetic hypothesis proposed by Fau & Villier (2020) based on morphological characters showed high congruence with Mah & Foltz (2011a). Six morphological synapomorphies were found for the clade Asteriidae (Fau & Villier, 2020): (i) a strongly arched abactinal profile of the ambulacrals, (ii) a long muscle insertion lim that finishes under the actam, (iii) the absence of round psas on the inferomarginals (spines attaching on keyhole-shaped psas, instead), (iv) abactinals differentiate with at least two distinct plate shapes, (v) the presence of intercalary inter-arc abactinals, and (vi) the presence of wreath organs. Only the presence of wreath organs is a non-ambiguous synapomorphy. In the literature, the early Jurassic F. gaveyi, M. spiniger, H. hettangiurnus and G. amplipapularia have all been considered, at some point, as members of the family Asteriidae (Blake, 1990; Blake, 2002). Our results no longer support them as members of the clade Asteriidae (Figs. 9 and 10).

Early Jurassic forcipulataceans

The three early Jurassic taxa F. gaveyi, H. hettangiurnus and G. amplipapularia, have been interpreted as Asteriidae, mostly due to their highly compressed ambulacrals and adambulacrals. Wright (1863–1880, p. 101) wrote about F. gaveyi: “The structure of the ambulacral skeleton, which is so admirably preserved in this fossil, removes all doubt as to its true generic position and affinities”. Blake (1990) compared H. hettangiurnus and G. amplipapularia to the Zoroasteridae and Asteriidae, stating that they are ‘intermediate in many ways” (p. 104), but still concluded that they were both of asteriid affinities. At the time of Blake’s (1990) publication, the family Asteriidae was not as restricted as it is today and comprised the subfamilies Stichasteridae and Pedicellasteridae. Therefore, Blake’s (1990) assumption of the phylogenetic position of H. hettangiurnus and G. amplipapularia is actually congruent with our results.

The phylogenetic positions of F. gaveyi, H. hettangiurnus and G. amplipapularia in the analysis suggest a new evolutionary history of the group. None of them possess any of the synapomorphies of the Asteriidae as proposed by Fau & Villier (2020). Instead, they exhibit a mix of plesiomorphic characters of the Forcipulatida and derived characters. Noticeable derived characters shared by F. gaveyi, H. hettangiurnus and G. amplipapularia are the absence of wings on the ambulacrals (character 35) and the high level of compression of the adambulacrals (character 42). F. gaveyi, H. hettangiurnus and G. amplipapularia possess the following plesiomorphic characters: (i) they have madreporites that are neither fused or imbricated with a primary interradial (character 114, not applicable in F. gaveyi), and (ii) they do not have any crossed pedicellariae. We cannot exclude the hypothesis that the absence of crossed pedicellariae could be a taphonomic bias, as crossed pedicellariae are, in general, smaller than straight pedicellariae. The absence of crossed pedicellariae could be a plesiomorphic characters within the Forcipulatacea, as they are also absent in the Zorocallida, or it could be a convergent loss, as crossed pedicellariae are present in other closely related Jurassic taxa, such as P. davidsoni, D. boehmi, and M. spiniger.

Psammaster davidsoni and Dermaster boehmi

The Middle Jurassic D. boehmi and the Late Jurassic P. davidsoni are found to be sister taxa, and to be part of a polytomy in the 50% MRC tree (Fig. 10) along with the Early Jurassic taxa. They share with the Early Jurassic taxa the following characters: numerous adambulacral spines (four spines per adambulacral; character 50) and numerous actinal rows (at least three actinal rows, character 56). Few extant forcipulatids possess more than three adambulacral spines (a notable exception is the asteriid genus Perissasterias, which possesses up to seven adambulacral spines). It is thus noticeable that all the Jurassic forcipulatids analyzed here possess three to four adambulacral spines, grossly arranged in a transverse row, which could be the plesiomorphic condition.

P. davidsoni and D. boehmi share with G. amplipapularia the following plesiomorphic characters: presence of secondary spines on the abactinal skeleton (characters 62, 70 and 93), and presence of only one row of abactinals between the superomarginals and the carinals (character 79). The presence of secondary spines is shared with Zorocallida. The presence of only one row of abactinals is a plesiomorphic character, shared with Labidiaster annulatus, Pedicellaster hypernotius and Zorocallida, and convergent in the stichasterid Neomorphaster forcipatus.

Argoviaster occultus and Marbleaster spiniger

The Jurassic A. occultus and M. spiniger are found to be with the Cretaceous Pegaster stichos higher in the tree, and belong without any doubt to the Forcipulatida. The close relationship of A. occultus, M. spiniger and P. stichos to the Forcipulatida is supported by characters 26 (average compression of the ambulacrals) and 28 (ambulacral arch slightly concave). Average compression of the ambulacrals is shared by all forcipulatid and G. amplipaluaria, H. hettangiurnus, F. gaveyi, D. boehmi, and P. davidsoni. No complete ambulacrals were visible, thus character 28 was scored (1) slightly arch, or (2) strongly arch for both A. occultus or M. spiniger, pending check on availability of better-preserved fossils. Both A. occultus and M. spiniger possess ambulacrals with small proximal tilting of the ambulacral’s head in the proximal directions (character 31). A small proximal tilting of the ambulacral’s head is a plesiomorphic character with most forcipulatid having rather pronounced proximal tilting.

Keyhole-shaped psas have been found on the actinals in M. spiniger (character 120) and are also found on the actinals and inferomarginals of extant asteriids and sitchasterids. Round psas are also present on the inferomarginals of stichasterids but absent in asteriids. The presence of a keyhole-shaped psas on the actinals of M. spiniger supports a derived position.

Of the Jurassic taxa, only A. occultus, M. spiniger, and D. boehmi possess both crossed and straight pedicellariae. Only straight pedicellariae have been observe in F. gaveyi, H. hettangiurnus and G. amplipapularia, and only crossed pedicellaria have been found in P. davidsoni. However, the absence of straight pedicellariae in P. davidsoni is likely to be a preservation bias (Fau et al., 2020). Except for Cretasterias reticulatus (Gale & Villier, 2013), there is no evidence of the presence of wreath organs in any of the fossil taxa included here, which is the only non-ambiguous synapomorphies of the Asteriidae. Wreath organs are a concentration of crossed pedicellariae around some spines with dedicated muscles allowing the wreath of pedicellariae to move up and down the spines (Lambert, De Vos & Jangoux, 1984). Thus, none of the Jurassic taxa can be assigned to crown Asteriidae.

Other Jurassic forcipulataceans and their evolutionary significance

Our phylogenetic analysis agrees with the suggestions of Gale (2011a) and Mah & Foltz (2011a) that early Jurassic “asteriids” are not true Asteriidae. Our investigation suggests that crown Asteriidae were not yet present by the Early Jurassic. Historically, many fossil forms were described as Asterias or synonyms of it during the late 19th and early 20th century, including Asteracanthion oolithicum Terquem & Jourdy, 1869 (Bathonian), Asterias ranvillensis Porte, 1927 (Bathonian), and Asterias delongchampsi Morière, 1878 (Oxfordian). Their assignment to the genus Asterias appears unlikely, in view of the phylogeny of the Forcipulatacea, and they need to be reappraised in the future. Asterias? dubium Whitfield, 1877 (Jurassic) is a species based on very poor material that do not allow for observation of morphological characters (Clark & Twitchell, 1915), and is unlikely to represent the genus Asterias either (Clark & Twitchell, 1915; Whitfield, 1877; Whitfield, 1880). Two additional Jurassic fossil forms have been recently interpreted as members of the Asteriidae, Savignasterias villieri Gale, 2011b from the Oxfordian of France and Polarasterias janusensis Rousseau & Gale, 2018 (in Rousseau, Gale & Thuy, 2018) from the Tithonian of central Spitsbergen. Although Savignasterias villieri and P. janusensis were not included in this analysis, absence of all crown Asteriidae synapomorphies, as outlined here, challenges these classifications. Savignasterias villieri is known only from isolated body wall ossicles, the shape of which leaves no doubt regarding its forcipulatid affinities. Keyhole-shaped psas are present in M. spiniger, the Asteriidae, the Stichasteridae and the genus Heliaster but they are missing in S. villieri (Gale, 2011b). No adambulacral, ambulacral or oral frame ossicles were described. The material remains too incomplete to assess the species’ phylogenetic position more clearly. Rousseau, Gale & Thuy (2018) discussed the affinities of P. janusensis and concluded that even though no crossed pedicellariae have been found and the unusual arrangement of the body wall skeleton for extant asteriids, P. janusensis should be considered an asteriid because of its relatively short terminal ossicles, the strongly quadriserial arrangement of the tube feet, the morphology of its oral ossicles and basal piece of straight pedicellariae. Although this character combination can be found in Asteriidae, they are also found in other forcipulatid groups such as the Stichasteridae or the polyphyletic family Pedicellasteridae, and none of the listed characters were found to be synapomorphies of the Asteriidae here or by Fau & Villier (2020). A phylogenetic reappraisal of both S. villieri and P. janusensis is still required but is beyond the scope of the current paper.

Blake & Guensburg (2016) reported a new fossil Stichasteridae from the Oxfordian Swift Formation of Montana, Atalopegaster gundersoni. Unfortunately, the fossil’s preservation does not permit a detailed description. Blake & Guensburg (2016) placed it within the family Stichasteridae stating: “Based on overall shape, ossicular expression, and fusion of the arms, Atalopegaster is aligned with Neomorphaster and the fossil genera Argoviaster Hess, 1972, and Pegaster Blake & Peterson, 1993, in the Stichasteridae sensu Mah & Foltz (2011a)” (Blake & Guensburg, 2016; p. 1161). Unfortunately, a more detailed investigation of the phylogenetic position of A. gundersoni is not easily attempted owing to the poor preservation of the limited number of specimens recovered to date.

Cretaceous and younger forcipulataceans taxa

Only six forcipulataceans species have been described from the Cretaceous to date, among which three belong to the Zorocallida: Protothyraster priscus De Loriol, 1874, Alkaidia sumralli Blake & Reid, 1998 and Alkaidia megaungula Ewin & Gale, 2020 (Fau & Villier, 2023). The three other forcipulataceans known to date are Afraster scalariformis Blake, Breton & Gofas, 1996, an assumed “pedicellasterid” from the Coniacian of Angola, the stichasterids Pegaster stichos Blake & Peterson, 1993 from the Campanian of the USA, and the asteriid Cretasterias reticulatus Gale & Villier, 2013 from the Maastrichtian of Morocco. The oldest extinct taxa known to date to present evidence of crossed pedicellariae arranged in wreath organs, an important synapomorphy of the Asteriidae, is C. reticulatus (Gale & Villier, 2013). However, keyhole-shaped psas are restricted to the actinals in C. reticulatus (Gale & Villier, 2013; pers. obs.), whereas the presence of keyhole-shaped psas on the inferomarginals only, is one of the synapomorphies of the extant Asteriidae. New observations of Afraster scalariformis have shown evidence for preserved pedicellariae, both straight and crossed pedicellariae, and keyhole-shaped psas on inferomarginals, questioning its systematic position among the family Pedicellasteridae (M Fau, 2024, pers. obs.). Investigation of the phylogenetic position of Cretaceous forcipulatacean sea stars is therefore needed to better understand the origin and diversification of the modern families.

Brisingids fossil record is almost nonexistent, with only one occurrence known to date, Hymenodiscus, from the Miocene of Japan (Yamaoka, 1987). Mah & Foltz (2011a) argued upon a late diversification of the Brisingida, because of their derived phylogenetic position and their relatively young fossil record. Morphology-based phylogenies usually fail at recognizing a derived position of the Brisingida within the Forcipulatacea (Gale, 2011a; Fau & Villier, 2020). A reappraisal of Cretaceous and Cenozoic forcipulataceans are therefore needed to understand the complex evolutionary history of this group.

Conclusion

We taxonomically reevaluated six fossil taxa, placed these species in a phylogenetic context, and constructed an evolutionary timeline for major diversification events in the history of the Forcipulatacea. Our results provide substantial evidence for a delayed origination of the family Asteriidae. The combined phylogenetic analysis of fossil and extant taxa suggests that the Jurassic forms exhibited characters that distinguish them from the extant families or genera they were previously assigned to and are characterized by unique combinations of plesiomorphic and derived characters. This also implies a progressive acquisition of characters leading to the extant crown group families after the Jurassic. None of the eight Jurassic species analyzed here were placed within any of the extant families, but instead represent parts of stem-groups. The clade comprising the Asteriidae and Stichasteridae has no fossil record before the Late Cretaceous. The absence of known Jurassic asteriids suggests a Cretaceous or even younger origin for the clade, and phylogenetic divergence dating provides evidence in favor of a delayed origination and diversification of this major clade. Understanding the timing and pace of diversification of the Asteriidae is of great interest to understanding recent biogeographical patterns, as emphasized by Mah & Foltz (2011a) and the undeniable evolutionary success of the family, which is the third largest family in terms of species diversity of all Neoasteroidea. Evolution of the crown-group characters during the Mesozoic was more progressive than formerly accepted. Moreover, our results contradict the idea of a rapid diversification of the Forcipulatacea during the Triassic or the earliest Jurassic.

Supplemental Information

Supplemental Information 1 Character/taxon matrix with tip-dating analysis script

Supplemental Information 2 List of characters

The authors are grateful to the collection managers who facilitated access to the specimens: Walter Etter, Sergio Kühni (NMB Basel); Guenter Schweigert (SMNS); Timothy A.M. Ewin (NHMUK); Christian Meister(MHNG); Marc Eléaume, Pierre Lozouet (MNHN); Eric A. Lazo-Wasem and Lourdes Rojas (YPM). MF thanks Gene Hunt, from the Paleobiology Department of the National Museum of Natural History; for acting as a sponsor during her fellowship appointment. We thank Thomas Saucède and two anonymous reviewers for their comments that helped improve the manuscript.

Institutional Abbreviations

NHMUK Natural History Museum, London, United Kingdom, formerly the British Museum of Natural History (BMNH).

NMB Naturhistorisches Museum Basel, Basel, Switzerland.

SMNS Staatliches Museum für Naturkunde Stuttgart, Stuttgart, Germany.

Additional Information and Declarations

Competing Interests

Author Contributions

Data Availability

New Species Registration

The authors declare there are no competing interests.

Marine Fau conceived and designed the experiments, performed the experiments, analyzed the data, prepared figures and/or tables, authored or reviewed drafts of the article, and approved the final draft.

David F. Wright performed the experiments, analyzed the data, authored or reviewed drafts of the article, and approved the final draft.

Timothy A.M. Ewin analyzed the data, authored or reviewed drafts of the article, and approved the final draft.

Andrew S. Gale analyzed the data, authored or reviewed drafts of the article, and approved the final draft.

Loïc Villier conceived and designed the experiments, performed the experiments, analyzed the data, prepared figures and/or tables, authored or reviewed drafts of the article, and approved the final draft.

The following information was supplied regarding data availability:

The character/taxon matrix and MrBayes script are available in the Supplementary File.

The following information was supplied regarding the registration of a newly described species:

Publication LSID: urn:lsid:zoobank.org:pub:6A43BD80-6C00-42C6-AFD4-7C4A944396FD

Forbesasterias: urn:lsid:zoobank.org:act:B14659DD-7EBD-411B-9471-54E6D9458288

Marbleaster: urn:lsid:zoobank.org:act:6CB3E261-2BA3-48DB-A17C-9E7B3586A431.

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
