# Peer review of "Phylogenetic and taxonomic revisions of Jurassic sea stars support a delayed evolutionary origin of the Asteriidae"

_PeerJ, doi:10.7717/peerj.18169_

## Round 0.1 · original submission · Minor Revisions

Dear authors, thank you very much for submit your research to PeerJ. Three researchers specialized in the topic have been reviewed your manuscript and they consider that minor changes will be necessary; please include a point by point letter where you discuss all the changes made. All the best and congratulations for your work

Reviewer 1 ·

Basic reporting

The sea star work of Fau and collaborators is an interesting and much needed study considering the fundamental uncertainties existing within the superorder Forcipulatacea. The dedicated focus on the highly diversified Asteriidae family is also welcome. The writing is clear and the English is good, the topic being well introduced. Overall, I do not have major concerns and I therefore think that the paper would be acceptable for publication in PeerJ after some minor revisions.

Experimental design

---Introduction---

- The introduction is well written, showing the meaningfulness of the work. The research questions are relevant and well-defined.

- L79: as there is no recent work suggesting the presence of only two families in the Forcipulatacea, I would write this paragraph differently. For example, by mentioning that two families were originally recognized (e.g. Fisher 1928), then that the systematics has highly changed until now where many families are now recognized. This should also be modified in the Abstract (L31).

---Material & Methods---

- This is a well undertaken study from a methodological point of view.

- L163: the inclusion of both extinct and extant species is a strength of your study. However, it’s unclear (if not looked in the supplementary material 1, which is not done for that purpose) to easily know which characters can be observed in both extinct and extant taxa. Could you make this clearer?

- L174: you mention the scoring up to 50% complete but there are definitely taxa with less scoring than this. Please also give average and minimum values.

Validity of the findings

---Systematic palaeontology---

- L381: wreath organ is an important character for your work, but its definition might be unknown to some readers. Please move the definition (L769) earlier in the manuscript.

- L395: Could you elaborate on why you decided to establish the new Marbleaster genus? Why are the keyhole-shaped psas and the peculiar pattern of pedicellariae key characters in your opinion?

---Results---

- I’d remove the references to previous works in this Results section and leave them for Discussion (e.g. “which is consistent with the results in Fau & Villier 2023”).

---Discussion---

- For posterior probabilities (PP), you used 0.5 as a threshold to merge into polytomies. This seems pretty low despite being expected for morphological characters. Please discuss this a bit further, compare it to other studies, and suggest whether/how this could be improved or not in the future.

- L853: do not have fossil a record -> do not have a fossil record

- L854: the absence of Jurassic asteriids suggest -> the absence of Jurassic asteriids suggests

- One thing that is missing in the discussion (and in the abstract) is a short synthesis on which families would the authors recognise in the Forcipulatacea (after their current study).

Additional comments

--- Tables and Figures ---

Figure 8: I find the characters harder to observe than on the other figures. I know this is partially related to the specimen but is there any way this could be improved?

·

Basic reporting

non comment

Experimental design

no comment

Validity of the findings

no comment

Additional comments

The ms by Fau and co-authors presents a dated morphology-based phylogeny of the superorder Forcipulatacea, a species rich and diversified group within the Asteroidea, including both extant and fossil taxa. Based on a fine description and interpretation of very well-preserved Jurassic material, they propose a taxonomic revision of Jurassic asteroids and describe two new genus: Forbesasterias gen. nov. and Marbleaster gen. nov. Using Bayesian inferences and a tip-dating analysis to date the phylogeny, their results support a more recent origination of modern families, and of Asteriidae in particular, than formerly assumed with Jurassic representatives forming a stem-group within the superorder Forcipulatacea. Their results also highlight an overlooked Jurassic diversity of the group.

I just have minor concerns of form (please see the joined annotated ms in pdf format).

Please be consistent when using either asteriid of Asteriidae, Trichasterospida or trichasterospids, and so on,…

At the end of the introduction, the authors should more clearly indicate that new genus are described.

The authors are very categorical in their conclusion writing that their results are an unequivocal demonstration of the delayed origin of the Asteriidae. I would be less positive, and a bit more prudent saying that the studied fossils do not support a Jurassic origin of the Asteriidae, and that their dated phylogeny suggests/supports a delayed origin of the family (as written in the title).

In the discussion / conclusion, the authors should better highlight the impact of their findings for our understanding of the evolution and biogeography of asteroids, which is not obvious for non specialists.

Figure 4C : as the difference between the two types of pedicellariae is not easy to see from the image only, a drawing or SEM image would help.

Figure 9. In the tree, is it worth keeping tree nodes with very low posterior probabilities (0. 12, 0.22, 0.28, and so on…) as this provides no information and may hinder the reading/understanding of the tree ?

Figure 9&10 : can the authors show on the trees, the known stratigraphic range of fossils (for instance, with dotted-lines or points) as if I am right, only FAD are used and shown in the tree ? For uninformed readers, branch length of fossil taxa might be read as true stratigraphic range.

Figure 9&10 : can the authors explicitly precise the meaning of black dots for identifying main sub-clades ?

Reviewer 3 ·

Basic reporting

no comment

Experimental design

no comment

Validity of the findings

no comment

Additional comments

There are only minor changes that need to be done: the most important typo would be the misspelling of Forbesasterias gen. nov. in the Systematic paleontology section.

References' format needs to be revised.

Annotated reviews are not available for download in order to protect the identity of reviewers who chose to remain anonymous.

---

## Round 0.2 · accepted · Accept

Dear authors of the paper, congratulations for the publication in the journal PeerJ, good research, I think the research will be a reference paper in the topic of fossil sea stars; all the 3 reviewers agree with the final version, all the best

Hugo

Reviewer 1 ·

Basic reporting

This revised version of the manuscript addresses nearly all my concerns from the initial version.

I notably appreciated the changes made in the Introduction and Material and Methods.

These were only minor requirements anyway and I consider that the paper can now be accepted in PeerJ in its current form.

Experimental design

see Basic reporting

Validity of the findings

see Basic reporting

·

Basic reporting

no comment.

Experimental design

no comment

Validity of the findings

no comment

Reviewer 3 ·

Basic reporting

The requested modifications have been successfully incorporated, resulting in a satisfactory outcome.

Experimental design

The requested modifications have been successfully incorporated, resulting in a satisfactory outcome.

Validity of the findings

The requested modifications have been successfully incorporated, resulting in a satisfactory outcome.